# Daphnane-Type Diterpenes from *Stelleropsis tianschanica* and Their Antitumor Activity

**DOI:** 10.3390/molecules27175701

**Published:** 2022-09-04

**Authors:** Xiaoyan He, Xiatiguli Abulizi, Xiaowan Li, Guoxu Ma, Zhaocui Sun, Hongyan Wei, Xudong Xu, Leiling Shi, Jing Zhang

**Affiliations:** 1College of Chinese Medicine Material, Jilin Agricultural University, Changchun 130118, China; 2Xinjiang Institute of Chinese and Ethnic Medicine, Urumqi 830002, China; 3Xinjiang Agricultural Vocational and Technical College Biological Technology Branch, Changji 831100, China; 4Institute of Medicinal Plant Development, Chinese Academy of Medical Sciences and Peking Union Medical College, Beijing 100193, China

**Keywords:** *Stelleropsis tianschanica*, daphnane-type diterpenoids, HGC-27 cells

## Abstract

Four new daphnane-type diterpenes named tianchaterpenes C-F (**1**–**4**) and six known ones were isolated from *Stelleropsis tianschanica*. Their structures were elucidated based on chemical and spectral analyses. The comparisons of calculated and experimental electronic circular dichroism (ECD) methods were used to determine the absolute configurations of new compounds. Additionally, compounds **1**–**10** were evaluated for their cytotoxic activities against HGC-27 cell lines; the results demonstrate that compound **2** had strong cytotoxic activities with IC_50_ values of 8.8 µM, for which activity was better than that of cisplatin (13.2 ± 0.67 µM).

## 1. Introduction

The morbidity and mortality of malignant tumors are the highest in many countries and regions worldwide, seriously threatening human life and health [1]. However, many new anticancer drugs are expensive; long-term applications can easily cause drug resistance and side effects [2]. Therefore, searching for high-efficiency and low-toxicity anticancer drugs is the key to treating tumors. Daphnane-type diterpenes are a kind of diterpene with a 5/7/6-tricyclic ring system. They are mainly distributed in plants of the *Thymelaeaceae* and *Euphorbiaceae* families [3]. Modern pharmacological studies have shown that daphnane-type diterpenes have antitumor [4], antiHIV [5,6], antineuroinflammatory, insecticidal, and other effects [7,8]. In recent years, many daphnane-type diterpenes have attracted the interest of researchers on account of their remarkable anticarcinogenic and antiHIV activities [9,10,11,12]. As a result, finding and screening new daphnane-type diterpenes as active anticancer components or lead compounds from herbal medicine has become a research hotspot in the natural product field [13].

*Stelleropsis tianschanica* is a member of the genus *Stelleropsis* in the *Thymelaeaceae* family [14]. It grows on the hillside grassland at an altitude of 1700–2000 m. It is only distributed in Zhaosu County, Xinjiang, China [15]. *S. tianschanica* has a certain genetic relationship with *Stellera chamaejasme* Linn and has many common components, which are used as a counterfeit of traditional Chinese medicine *Stellera chamaejasme* in Xinjiang [16]. Due to the unique geographical environment and growth area, there is little research on the functional components of *S. tianschanica*. Our team found that *S. tianschanica* contained novel daphnane-type diterpenes (Appendix A) [17]. To further -obtain potential daphnane-type diterpenes from *S. tianschanica*, its methanol, CH_2_Cl_2_, and EtOAc extractions were investigated, and ten daphnane-type diterpenes (Figure 1), including four new ones (**1**–**4**) and six known compounds (**5**–**10**), were obtained. Among them, compound **1** possessed a 5/7/6/3-tetracyclic skeleton, which is not common in daphnane-type diterpenes. In addition, the antitumor activity of the isolated compounds was tested. In this paper, we report the structure elucidation and biological screening in vitro of the obtained daphnane-type diterpenes from *S. tianschanica*.

## 2. Results

### 2.1. Structure Elucidation

Compound **1**, a white amorphous powder, showed 12 degrees of unsaturation based on the molecular formula (C_27_H_32_O_8_), which was determined by the HR-ESI-MS ion at *m*/*z* 507.1974 [M + Na] ^+^ (calcd. for C_27_H_32_O_8_Na, 507.1989). In the ^1^H NMR (600 MHz, CDCl_3_) spectrum (Table 1), compound **1** exhibited characteristic proton signals assignable to a gem-dimethyl cyclopropane moiety at *δ*_H_ 1.25 (1H, d, *J* = 7.2 Hz, H-14), 1.15 (3H, s, H-16), and 1.31 (3H, s, H-17). The abovementioned gem-dimethyl cyclopropane moiety proton resonances, together with the two methyl group signals at *δ*_H_ 0.95 (3H, d, *J* = 6.6 Hz, H-18) and 1.79 (3H, dd, *J* = 1.8, 2.4 Hz, H-19), indicated that compound **1** contained a tigliane skeleton [18]. Additionally, *α*, *β*-unsaturated carbonyl groups were proposed in compound **1** based on the observation of downfield shifted olefinic data at *δ*_H_ 7.75 (1H, s, H-1) [9,19], *δ*_C_ 164.5 (C-1), *δ*_C_ 134.0 (C-2), and a conjugated keto carbonyl carbon signal at *δ*_C_ 210.2 (C-3). Three aromatic protons at *δ*_H_ 8.05 (1H, dd, *J* = 1.2, 8.4 Hz), 7.60 (1H, t, *J* = 7.2 Hz), and 7.46 (1H, t, *J* = 7.8 Hz) suggested the presence of a benzene moiety. Meanwhile, there were two oxygenated protons at *δ*_H_ 4.27 (1H, s, H-5) and 3.30 (1H, s, H-7) in the structure. The ^13^C NMR spectrum displayed 32 carbon signals, among which there was a carbonyl carbon on the benzoyl group *δ*_C_ 167.9 (C-1′); epoxy groups *δ*_C_ 61.8 (C-6) and *δ*_C_ 65.8 (C-7); and four methyl signals at *δ*_C_15.8 (C-16), 23.0 (C-17), 19.1 (C-18), and 9.86 (C-19). In fact, the NMR data of compound **1** were similar to those reported of 6*α*,7*α*-epoxy-5*β*-hydroxy-12-deoxyphorbol-13-tetradecanoate, except for the benzoyl group linked on *δ*_C_ 64.7 (C-13) [18]. The proton signals were assigned to the corresponding carbons through direct ^1^H and ^13^C correlations in the HSQC spectrum. From the ^1^H-^1^H COSY analysis, four substructures (drawn with bold bonds in Figure 2) were established as H-1/H-10/H_3_-19, H-8/H-14, H-11/H-12, and H-11/H_3_-18. In the HMBC spectrum (Figure 2), the correlations of H-18 (*δ*_H_ 0.95) with C-9 (*δ*_C_ 75.4), C-12 (*δ*_C_ 32.0), and H-19 (*δ*_H_ 1.79) with C-1 (*δ*_C_ 164.5) and C-3 (*δ*_C_ 210.2) suggested that two methyl groups were linked to C-2 and C-11, respectively. In addition, H-16/C-15, H-16/ C-13, H-17/C-15, H-17/C-13, and H-8/C-15 confirmed that compound **1** was a tigliane-type diterpene, which is not common in daphnane-type diterpenes. In the NOESY spectrum (Figure 3), there was enhancement between H-5 and H-10 and H-7 and H-8; however, there was no correlation between H-8 and H-14. This suggested the *β*-orientation of 5-OH and *α*-orientation of 13-benzoyl. All the daphnane-type diterpenes isolated to date from the medicine plant have the same carbon skeleton with the *trans*/*trans* system of the three rings, A, B, and C, *β*-oriented for H-8 and the hydroxyl group at C-5, and they are *α*-oriented for H-10 and the hydroxyl group at C-9. This is well-established. A quantum chemical electronic circular dichroism (ECD) calculation was performed using the TDDFT methodology at the b3lyp/6-311 + g (d, p) level in MeOH to confirm its absolute configurations, in which the TDDFT-calculated ECD curve matched well with the experimental ECD spectrum (Appendix A). Thus, the structure of compound **1** was determined, shown as and named tianchaterpene C.

Compound **2** was obtained as a white powder with its molecular formula assigned as C_32_H_42_O_10_ according to the positive HR-ESI-MS peak at *m*/*z* [M + Na] ^+^ 609.2652 (calcd. for C_32_H_42_O_10_Na, 609.2670), exhibiting eleven degrees of unsaturation. Through the ^1^H- and ^13^C-NMR spectra, we inferred that the basic nucleus of compound **2** was a daphnane-type diterpene, which was further confirmed by the ^1^H–^1^H COSY and HMBC spectra (Figure 2). Meanwhile, in the HMBC spectrum, the correlations from *δ*_H_ 5.01 (1H, s, H-12) to *δ*_C_ 169.4 (C-1’) indicate that the acetoxyl group is attached to C-12, and the correlations of H-14 (*δ*_H_ 4.83), H-2″ (*δ*_H_ 5.59, d, *J* = 15 Hz), and H-3″ (*δ*_H_ 6.63, dd, *J* = 10.8, 15 Hz) with C-1″ (*δ*_C_ 116.8) suggested that the adipose chain was linked to C-1″. Based on the olefinic proton signals with large coupling constants in the ^1^H NMR data, the fatty group was assigned as *trans*-conjugated diene. In fact, the NMR data of compound **2** were similar to those reported for yuanhuadine [20,21,22], except for the different orientation of the epoxy group at *δ*_C_ 75.2 (C-6) and 80.1 (C-7), which is located in the low field area compared with yuanhuadine. In the NOESY spectrum, the correlations of H-5/H-10, H-5/H-7, and H-7/H-20 indicated *α*-orientation for these protons. Therefore, it is judged that the epoxy group has *β*-orientation, while the yuanhuadine group has *α*-orientation. In conclusion, the structure of compound **2** was determined as the isomer of yuanhuadine. The calculated and experimental ECD data also supported this deduction. Accordingly, the structure of compound **2** was established as shown and named tianchaterpene D.

Compound **3** was a white amorphous powder. The molecular formula C_30_H_40_O_9_ was established from the HR-ESI-MS spectrum with a positive ion peak at *m*/*z* [M + Na] ^+^ 567.2545 (calcd. for C_30_H_40_O_9_Na, 567.2565). The ^1^H NMR spectrum revealed two olefinic protons at *δ*_H_ 5.08 (1H, s, H-16) and 5.11 (1H, s, H-16) and five methyl groups at *δ*_H_ 0.87 (3H, t, H-10′′), 1.88 (3H, s, H-17), 1.23 (3H, d, *J* = 7.2, H-18), and 1.80 (3H, s, H-19), as shown in Table 1. The ^13^C NMR spectrum indicated the presence of epoxy groups *δ*_C_ 60.6 (C-6) and 64.5 (C-7); *α*, *β*- unsaturated cyclopentanones *δ*_C_ 210.0 (C-3), 136.9 (C-2), and 160.9 (C-1); and ortho ester groups 78.7 (C-9), 85.3 (C-13), 81.0 (C-14), and 116.9 (C-1′′). All the NMR signals were typical for a daphnane-type diterpene as compound **3** was further confirmed by HMBC correlations of H-16/C-17, H-16/C-13, H-12/C-15, H-12/C-9, H-5/C-7, H-14/C-7, H-14/C-1′′, and H-2′′/C-1′′ (Figure 2). The similarity of the NMR signals of compound **3** to those of 12-hydroxydaphnetoxin [21], except for the conjugated diene signals (H-2′′ to H-5′′), suggested that the two compounds are geometric isomers. Based on the coupling constant (*J*) in the ^1^H NMR spectrum (*J*_H-2′′,H3′′_ = 15.6 Hz, *J*_H-4′′,H-5′′_ = 11.4 Hz), compound **3** contains a *trans*, *cis*-conjugated diene, while 12-hydroxydaphnetoxin contains a *trans*, *trans*-conjugated diene unit. The correlations H-2′′ to H-10′′ in the ^1^H-^1^H COSY spectrum indicated that the aliphatic group is a straight chain. Its location at C-14 was supported by the HMBC correlation between H-14/C-1′′. In the NOESY spectrum, the correlation between H-18/H-12 indicated that the protons are in the same plane. In addition, it revealed that H-11 and H-12 have a trans-orientation, which meant *β*-orientation of 12-OH. The correlations of H-8/H-11, H-8/H-14, H-8/H-7, and H-7/H-14 indicated *β*-orientation for these protons, while the enhancement between H-5/H-10 suggested *α*-orientation for H-5 and H-10. Taken together with its calculated and experimental ECD spectra, the structure of compound **3** was determined as shown and named tianchaterpene E.

Compound **4**, purified as a white powder, has a molecular formula of C_37_H_44_O_10_, deduced from the HR-ESI-MS quasimolecular ion at *m*/*z* 671.2830 [M + Na] ^+^ (calculated C_37_H_44_O_10_Na_,_ 671.2827). The ^1^H NMR and ^13^C NMR spectroscopic data (Table 1) of **4** were similar to those of yuanhuacine. However, based on the coupling constant (*J*) in the ^1^H NMR spectrum, the conjugated diene signals (H-2′′ to H-5′′) contain a *trans*, *cis*-conjugated diene, while yuanhuacine contains a trans, trans-conjugated diene unit, suggesting that the two compounds are geometric isomers. The correlations H-2′′ to H-10′′ in the ^1^H-^1^H COSY spectrum indicated that the side aliphatic group is a straight chain. In the HMBC spectrum (Figure 2), the correlations from *δ*_H_ 5.21 (1H, s, H-12) to *δ*_C_ 165.4 (C-1’) suggested that the methyl benzoate group is attached to C-12. In the NOESY spectrum, correlations were observed among H-12/H-18, H-8/H-11, H-8/H-14, H-7/H-14, and H-5/H-10, which were analogous to those for compound **3,** as supported by their identical ECD spectra (Appendix A). As a result, compound **4** was elucidated as shown and named tianchaterpene F.

### 2.2. Antitumor Activity

Compounds **1**–**10** isolated from the *Stelleropsis tianschanica* were measured for their antitumor activities against HGC-27 human gastric cancer cells, with cisplatin as the positive control (Table 2). Among them, compound **2**, which contain a 6,7-epoxide group of β-orientation, showed stronger inhibitory activity than the other compounds, indicating that the group may enhance the cytotoxic activity against HGC-27 gastric cancer cells, with an IC_50_ of 8.8 µM. In addition, compounds **8**–**10** (IC_50_ ≤ 17.5 µM), which each had an orthoester group and a decadiene group, showed stronger activity than all compounds except **6**, suggesting that the orthoester and decadiene groups are both significant in the inhibitory activities against HGC-27 cells. This result is consistent with previous studies on daphnane-type diterpenoids [23]. At the same time, the experimental results also show that compound **2** had the best inhibitory effect; its activity was better than that of cisplatin, and the activity of tigliane skeleton diterpenes was lower than that of daphnane-type diterpenes.

## 3. Materials and Methods

### 3.1. General Experimental Procedures

Optical rotation data were measured using a Perkin-Elmer 341 digital polarimeter (PerkinElmer, Norwalk, CT, USA). UV and IR spectral data were recorded on Shimadzu UV2550 and FTIR-8400 spectrometers (Shimadzu, Kyoto, Japan). CD spectra were obtained using a JASCO J-815 spectropolarimeter. NMR spectra were obtained using a Bruker AV III 600 NMR spectrometer with chemical shift values presented as *δ* values using TMS as the internal standard (Bruker, Billerica Germany). HR-ESI-MS was performed using an LTQ-Orbitrap XL spectrometer (Thermo Fisher Scientific, Boston, MA, USA); samples were dissolved in chromatographic methanol and treated through a membrane, single pump. Semipreparative HPLC was performed using an HPLC PUMP K-501, LC3000 high-performance liquid chromatograph (Beijing Tong Heng Innovation Technology Co., Ltd., Beijing, China), and Kromasil 100-5C18, 250 × 10 mm, E108850. Column chromatography (CC) was performed using silica gel (100–200 and 200–300 mesh, Qingdao Marine Chemical Plant, Qingdao, China). Precoated silica gel GF254 plates (Zhi Fu Huang Wu Pilot Plant of Silica Gel Development, Yantai, China) were used for TLC. All solvents used petroleum ether, ethyl acetate, dichloromethane, methanol (analytical grade and chromatographic grade), and deuterated chloroform that were of analytical grade (Beijing Chemical Plant, Beijing, China).

### 3.2. Plant Material

The *Stelleropsis tianschanica* Pobed. were collected in October 2020 from Zhaosu City, Xinjiang Autonomous Region, China and authenticated by Prof. Xiaoguang Jia. A voucher specimen (20201022) was deposited at the Xinjiang Institute of Chinese and Ethnic Medicine.

### 3.3. Isolation and Purification of Compounds ***1***–***10***

Dried powders of *Stelleropsis tianschanica* roots (4.0 kg) were extracted three times with methanol under reflux. Removal of the methanol under reduced pressure yielded the methanol extract (684.0 g). The methanol extract was dissolved in water and extracted three times with dichloromethane (3 × 1000 mL). The dichloromethane fraction (203.0 g) was subjected to CC over a silica gel (100–200 mesh), eluting with a stepwise gradient of CH2Cl_2_/MeOH (from 1:0 to 0:1; i.e., 1:0, 100:1, 30:1, 20:1, 10:1, 5:1, 2:1, and 0:1, *v*/*v*) to yield fractions A–H. Fr.D was subjected to CC over a silica gel (100–200 mesh), eluting with a stepwise gradient of petroleum ether/EtOAC (from 5:1 to 0:1, i.e., 5:1, 3:1, 2:1, 1:1, and 0:1, *v*/*v*) to yield five fractions (Fr.D 1–5). Fr.D 3 was isolated through ODS MPLC elution with MeOH/H_2_O (50:50, 60:40, 70:30, 80:20, 90:10, and 100:0, *v*/*v*) to yield six fractions (Fr.D 3-1–6). Fr.D 3-3 was purified using semipreparative HPLC with MeOH/H_2_O (70:30, *v*/*v*) as the mobile phase to yield compound **1** (2.3 mg, *t*_R_ = 54.3 min); **5,** Phorbol 12-benzoate 13-acetate 20-homovanillate (2.3 mg, *t*_R_ = 33.4min); **6,** yuanhuaoate A (1.8 mg, *t*_R_ = 42.2 min); and **7,** vesiculosin (3.7 mg, *t*_R_ = 61.3 min). Fr.D 3-5 was purified using semipreparative HPLC with MeOH/H_2_O (85:15, *v*/*v*) as the mobile phase to yield compound **2** (4.1 mg, *t*_R_ = 36.7 min); **8,** yuanhuacine (2.7 mg, *t*_R_ = 58.6 min); and **9,** daphgenkin A (3.1 mg, *t*_R_ = 62.1 min). Fr.D 3-4 was isolated through gel HW-40C elution with MeOH to yield two fractions (Fr.D 3-4-1–2). Fr.D 3-4-2 was purified using semipreparative HPLC with MeOH/H_2_O (80:20, *v*/*v*) as the mobile phase to yield compound **3** (2.4mg, *t*_R_ = 41.2 min). Fr.C was subjected to CC over a silica gel (100–200 mesh), eluting with a stepwise gradient of CH_2_Cl_2_/MeOH (from 50:1 to 0:1, i.e., 50:1, 30:1, 10:1, 5:1, and 0:1, *v*/*v*) to yield five fractions (Fr.C 1–5). Fr.C 2 was isolated through ODS MPLC elution with MeOH/H_2_O (50:50, 60:40, 70:30, 80:20, 90:10, and 100:0, *v*/*v*) to yield six fractions (Fr.C 2-1–6). Fr.C 2-5 was isolated through gel HW-40C elution with MeOH to yield three fractions (Fr.C 2-5-1–3). Fr.C2-5-3 was purified using semipreparative HPLC with MeOH/H_2_O (85:15, *v*/*v*) as the mobile phase to yield compound **4** (1.7 mg, *t*_R_ = 47.9 min). Fr.C 2-4 was purified using semipreparative HPLC with MeOH/H_2_O (77:23, *v*/*v*) as the mobile phase to yield compound **10,** yuanhuadine (3.5 mg, *t*_R_ = 29.2 min).

### 3.4. Characterization of Compounds ***1***–***4***

Tianchaterpene C (**1**): white powder (MeOH); ${\left[\alpha \right]}_{D}^{25}$ −20.0 (*c* 0.025, MeOH); UV (MeOH) (log *ε*) λ_max_ 234 (2.95) nm; IR (film) ν_max_ 3443, 2925,1632, 1384, 1287, 1114 cm^−1^; ECD (MeOH) λ_max_ (Δ*ε*) 213.3 (+29.24), 228.7 (−5.07), 240.9 (+3.7), 272.6 (−18.7) nm; ^1^H-NMR and ^13^C-NMR data (CDCl_3_), see Table 1; HR-ESI-MS *m*/*z* 507.1974 [M + Na] ^+^ (calculated 507.1989, C_27_H_32_O_8_Na).

Tianchaterpene D (**2**): white powder (MeOH); ${\left[\alpha \right]}_{D}^{25}$ +12.0 (*c* 0.025, MeOH); UV (MeOH) (log *ε*) λ_max_ 232 (2.92) nm; IR (film) ν_max_ 3436, 2927, 2856, 1740, 1632, 1454, 1377, 1036 cm^−1^; ECD (MeOH) λ _max_ (Δ*ε*) 217.6 (+39.8), 230.2 (+27.4), 258.3 (−59.9) nm; ^1^H-NMR and ^13^C-NMR data (CDCl_3_), see Table 1; HR-ESI-MS *m*/*z* 609.2652 [M + Na] ^+^ (calculated 609.2670, C_32_H_42_O_10_Na).

Tianchaterpene E (**3**): white powder (MeOH); ${\left[\alpha \right]}_{D}^{25}$ +24.00 (*c* 0.025, MeOH); UV (MeOH) (log *ε*) λ_max_ 234 (2.86) nm; IR (film) ν_max_ 3430, 2926, 2857, 1700, 1694, 1630, 1453, 1382, 1318, 1031 cm^−1^; ECD (MeOH) λ _max_ (Δ*ε*) 251.4 (+31.5), 288.5 (−19.8), 350.6 (+12.8) nm; ^1^H-NMR and ^13^C-NMR data (CDCl_3_), see Table 1; HR-ESI-MS *m*/*z* 567.2545 [M + Na] ^+^ (calculated 567.2565, C_30_H_40_O_9_Na).

Tianchaterpene F (**4**): white powder (MeOH); ${\left[\alpha \right]}_{D}^{25}$ +71.99 (*c* 0.025, MeOH); UV (MeOH) (log *ε*) λ_max_ 233 (2.90) nm; IR (film) ν_max_ 3440, 2926, 2855, 1710, 1632, 1515, 1452, 1383, 1270 cm^−1^; ECD (MeOH) λ _max_ (Δ*ε*) 247.8 (+51.6), 275.2 (−77.1) nm; ^1^H-NMR and ^13^C-NMR data (CDCl_3_), see Table 1; HR-ESI-MS *m*/*z* 671.2830 [M + Na] ^+^ (calculated 671.2827, C_37_H_44_O_10_Na).

### 3.5. Antitumor Activity

The cytotoxic activities of compounds **1**–**10** against HGC-27 cells were tested using the MTT colorimetric method. HGC-27 cells were cultivated on DMEM medium at 37℃ and 5% CO_2_. After diluting the DMEM medium, cells were seeded into 96-well sterile microplates (4 × 10^5^ cells/well) and cultured with various concentrations of tested compounds or DDP (positive control) for 24 h at 37 °C. After incubation, all compounds were tested at five concentrations (10−100 µM) for 1 h. Following this, the supernatant was removed, and all components were dissolved in 100% DMSO at such an amount that there was a final DMSO concentration of 0.1% added to each well. The absorbance was measured using a microplate reader at a wavelength of 570 nm. Data are displayed as the means ± SD (*n* = 3). The cell growth assay was repeated three times, and the IC_50_ values were calculated using Microsoft Excel software.

## 4. Conclusions

Ten diterpenoids, including four new ones, were obtained from *Stelleropsis tianschanica*, and the complete structures of all of these new compounds were eventually elucidated by extensive spectral analysis. The antitumor activities of all isolates were tested. The results show that compounds **2** and **8**–**10** displayed potential biological activities, with IC_50_ values less than 20 µM, thus proving that daphnane-type diterpenes in *Stelleropsis tianschanica* can be used as potential antitumor drugs, which is worthy of attention. This manuscript also provides a theoretical basis for further clinical application, development, and utilization of the medicinal plant *Stelleropsis tianschanica*.

## Figures and Tables

**Figure 1 molecules-27-05701-f001:**
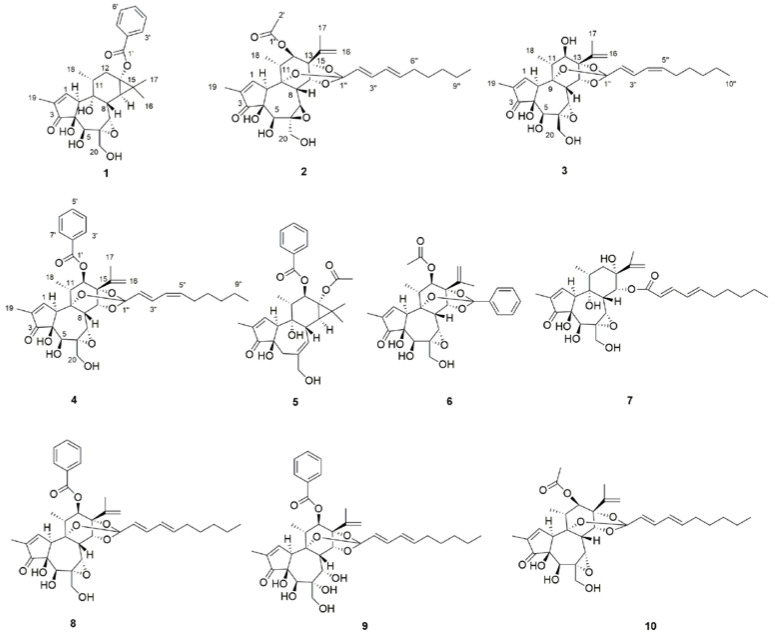
Structures of compounds **1**–**10,** compounds **1**–**4** are new compounds and **5**–**10** are known ones.

**Figure 2 molecules-27-05701-f002:**
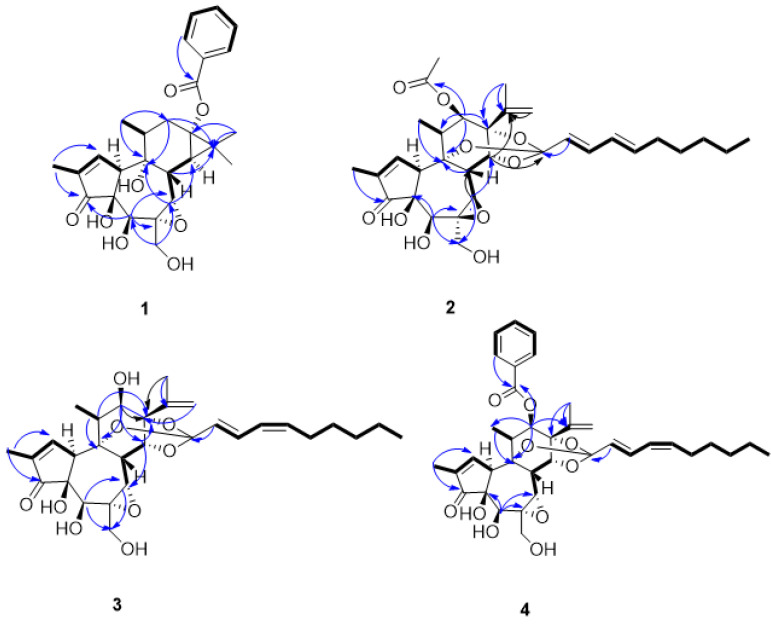
Key ^1^H-^1^H COSY (in bold) and HMBC (arrows) correlations of compounds **1**–**4**.

**Figure 3 molecules-27-05701-f003:**
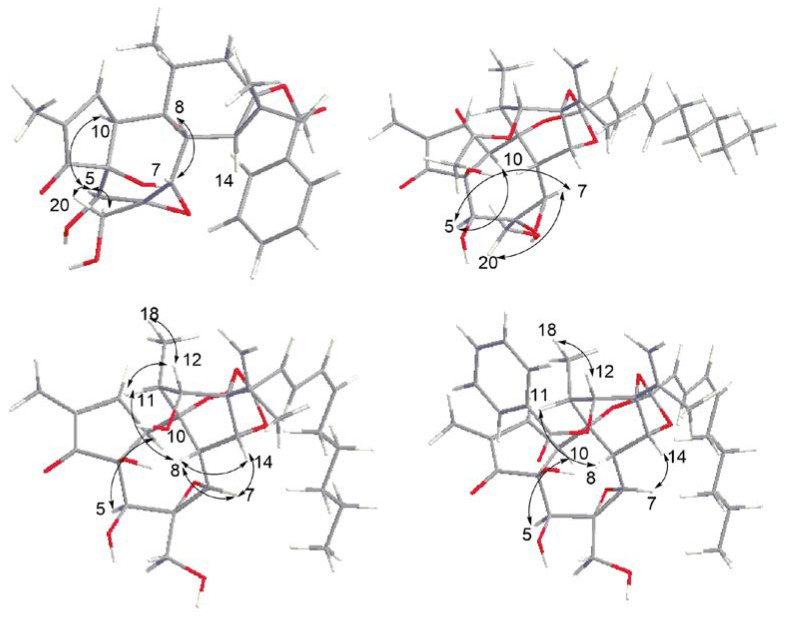
Key NOESY correlations of compounds **1**–**4**.

**Table 1 molecules-27-05701-t001:** NMR spectral data of **1**–**4** (600 MHz for ^1^H NMR and 150MHz for ^13^C NMR).

No.	1 *^a^*		2 *^a^*		3 *^a^*		4 *^a^*	
	*δ*_H_ (*J* in Hz)	*δ*_C,_ type	*δ*_H_ (*J* in Hz)	*δ* _C_	*δ*_H_ (*J* in Hz)	*δ* _C_	*δ*_H_ (*J* in Hz)	*δ* _C_
1	7.75, s	164.5	7.61, s	159.9	7.57, s	160.9	7.59, s	160.3
2	-	134.0	-	137.1	-	136.9	-	137.0
3	-	210.2	-	208.9	-	210.0	-	209.4
4	-	72.5	-	76.3	-	72.4	-	72.1
5	4.27, s	71.7	4.14, s	73.7	4.25, s	72.2	4.22, s	71.9
6	-	61.8	-	75.2	-	60.6	-	64.7
7	3.30, s	65.8	4.41, s	80.1	3.52, s	64.5	3.91, d, (7.2)	64.0
8	2.90, d, (7.2)	36.4	3.14, d, (2.4)	35.6	3.73, d, (2.4)	47.7	3.64, d, (2.4)	35.7
9	-	75.4	-	78.6	-	78.7	-	78.2
10	3.99, s	49.3	3.86, t, (2.4)	50.5	3.83, t, (2.4)	35.1	3.84, t, (2.4)	47.4
11	1.88, m	39.6	2.71, q, (7.2)	43.6	2.51, d, (7.8)	44.8	2.57, q, (7.2)	44.1
12	2.17, m	32.0	5.01, s	77.5	3.89, s	77.3	5.21, s	78.9
13	-	64.7	-	84.4	-	85.3	-	83.9
14	1.29, d,(7.2)	32.2	4.83, d, (2.4)	82.5	4.74, d, (2.4)	81.0	4.90, d, (2.4)	80.5
15	-	24.2	-	142.7	-	145.1	-	142.9
16	1.15, s	15.8	4.97, s, 5.02, s	113.5	5.08, s; 5.11, s	113.0	5.01, d, (7.2)	113.7
17	1.31, s	23.0	1.82, s	17.9	1.88, s	18.9	1.87, s	18.8
18	0.95, d, (6.6)	19.1	1.25, s	18.6	1.23, d, (7.2)	19.1	1.40, s	18.4
19	1.79, s	9.86	1.82, s	10.0	1.80, s	10.2	1.77, d, (1.8)	9.9
20	3.85, q	64.7	4.08, d, (11.4); 3.70, d,(11.4)	70.0	3.79, dd, (12.6)	65.2	3.94, t, (3.0)	60.5
1’	-	167.9	-	169.4	-	-	-	165.4
2’	-	129.7	1.96, s	21.0	-	-	-	129.6
3’	8.05, dd, (1.2, 8.4)	129.8	-	-	-	-	7.90, d, (7.2)	129.5
4’	7.46, t, (7.8)	128.6	-	-	-	-	7.46, t, (7.2)	128.6
5’	7.60, t, (7.2)	133.6	-	-	-	-	7.58, t, (5.4)	133.4
6’	7.46, t, (7.8)	128.6	-	-	-	-	7.46, t, (7.2)	128.6
7’	8.05, dd, (1.2, 8.4)	129.8	-	-	-	-	7.90, d, (7.2)	129.5
1″	-	-	-	116.8	-	116.9	-	117.1
2″	-	-	5.59, d, (15)	121.7	5.73, d, (15.6)	136.4	5.76, d, (15)	136.6
3″	-	-	6.63, dd, (10.8, 15)	135.3	6.95, dd, (11.4,15.6)	129.7	7.01, dd, (4.2,15.6)	129.8
4″	-	-	6.05, dd, (10.8, 15)	128.2	5.98, t, (11.4)	127.1	6.02, q, (11.4)	126.8
5″	-	-	5.89, dd, (7.2,15)	140.0	5.59, dd, (7.2,11.4)	124.9	5.61, dd, (4.8, 7.8)	124.2
6″	-	-	2.12, dd (6.6, 13.8)	32.6	2.19, m	28.0	2.22, q, (6.6)	27.9
7″	-	-	1.39, m	28.6	1.38, m	29.3	1.30–1.34, m	29.1
8″	-	-	1.27–1.30, m	31.3	1.23, m	31.6	1.30–1.34, m	31.4
9″	-	-	1.27–1.30, m	22.5	1.34, m	22.7	1.30–1.34, m	22.5
10″	-	-	0.89, t, (7.2)	14.0	0.87, t, (7.2)	14.3	0.88, t, (7.2)	14.1

*^a^* Spectra data were recorded in CDCl_3_.

**Table 2 molecules-27-05701-t002:** In vitro cytotoxic activities of compounds **1**–**10**.

Compounds	HGC-27	
	Inhibition Rate ^*c*^	IC_50_(µM)
Blank group	35.19 ± 0.69	-
**1**	67.26 ± 1.43	39.3 ± 1.35 *^a^*
**2**	93.22 ± 0.98 *	8.8 ± 0.81
**3**	90.89 ± 0.74	21.5 ± 0.72
**4**	92.55 ± 1.65	26.5 ± 1.02
**5**	80.85 ± 2.18	34.5 ± 0.68
**6**	68.66 ± 1.23	34.2 ± 0.97
**7**	69.03 ± 0.94	35.6 ± 1.16
**8**	93.19 ± 1.37 *	14.0 ± 0.24
**9**	92.52 ± 2.11	17.5 ± 0.43
**10**	94.92 ± 1.09 *	11.3 ± 0.99
Cisplatin *^b^*	93.85 ± 2.07	13.2 ± 0.67

*^a^* Value present mean ± SD of triplicate experiments; *^b^* Positive control substance; *^c^* Inhibitory rate of all compounds on HGC-27 cell lines at 50 µM; * *p* < 0.05.

## Data Availability

The data presented in this study are available in the Appendix A.

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
