# Peer review of "Daphnane-Type Diterpenes from Stelleropsis tianschanica and Their Antitumor Activity"

_molecules, 2022, doi:10.3390/molecules27175701_

Round 1

Reviewer 1 Report

Overall, the authors describe the isolation and structural determination of four new diterpenes tianchaterpenes C-F (1-4) along with six other known compounds from Stelleropsis tianschanica. The structures of new compounds 1-4 were elucidated on the basis of NMR and MS spectroscopic data analyses, as well as ECD method for the absolute configuration.  The authors also have evaluated the isolated compounds for their cytotoxic activities against HGC-27 cell lines. The compounds are well characterized for the most part, however there are some major revision/edits and concerns related to the manuscript/data which need to be revised prior to publication as summarized below: 

Comments/edits to authors:

-          In Title and throughout the manuscript: “Daphane ….” revise as “Daphnane-type ….”

-          Lines 32-33: “…some of them have been employed in a range of clinical applications for a variety of clinical uses [9] ….”, I did not see any relation between reference 9 and clinical uses of the class – please check?

-          Figure 1: The authors should be consistent with structure drawing (e.g. structure 10, the acetyl-group at 12-position was drawn in different formats as -OCOCH3?)

-          Line 58: “calcd for C27H32O8” revise as “calcd for C27H32O8Na” - sodium “Na” should be added to the calculated molecular formula throughout the manuscript.

-          The authors need to explain - why they used the name tianchaterpene C-F for compounds 1-4, I would recommend including the chemical structures of previously reported compounds tianchaterpene A-B in supporting information file?

-          Compounds numbers should be in “bold” throughout the manuscript.

-          ECD: The experimental data “values” for ECD of compounds 1-4 should be added in the experimental part.

-          Lines 95-96, lines 113, and lines 136-137: the formula for calcd HRMS is missing.

-          Line 97: delete the word “mother”

-          Line 104: the authors should add the reference for yuanhuadine?

-          Line 137: 1H- NMR and 13C (numbers should be superscript throughout the manuscript”

-          Line 172:  Optical rotation? I did not see any Optical rotation data in the manuscript. The optical rotation data for compounds 1-4 should be added in the experimental part, page 8 (lines 223-237).

-          Sodium (Na) should be added to the calculated molecular formula:  For example

Line 233: “C30H40O9” revise as “C30H40O9Na”

-          Table 1: The NMR data for some signals may need some revision: e.g. H-1 show as singlet (s) and based on structures it should be as doublet “d”, particularly the authors show the COSY correlations between H-1/H-10 (Figure 2)? Please explain. Also, the coupling constants for the H-3” (compound 2) is not matched with the trans? As shown in the structure 2. Same thing for the coupling constants of H-4” in compound 2, does not match with trans-coupling? Please revise for all structures 2-4 and check the data for side chain 1”-position to confirm the trans/cis of the double-bonds in the side chain.

-          Figure 3: The NOESY correlation/structures figure has very poor quality? And authors should also include the atom-numbers for this structure’s figures?

-          The figures of cytotoxicity data (against HGC-27) for the tested compounds 1-10 should be included in Supporting Information file. The authors show only the IC50 data in Table 2, but it is important to also include the figures of the cytotoxicity data.

-          Experimental part: Lines 194-221: The names of the other known compounds 5-10 should be included.

-          Optical rotation and ECD data should be added in the experimental part, page 8 (lines 223-237).

-          Lines 223-237: HRMS of compounds 1-4, sodium “Na” should be added to all the Calculated molecular formulae.

-          All ECD spectra of compounds 1-4 should be added in Supporting Information file or put all the four figures in main manuscript for better comparison.

-          Figure S26: The ECD spectra of calculated/Experimental data of compound 3 does not match well, this may be due to the low purity sample of 3 as mention previously (NMR spectra of compound 3). Please explain in the manuscript.

-          References should be revised according to the journal of “Molecules” formats, some journals are abbreviated, and some are not, etc. (e.g. ref 14, 17 are not abbreviated, etc.)

-          I would recommend that: The manuscript English language need to be revised for typographic spelling and grammatic errors and need through language editing.

Supporting Information:

-          In all figure legends and titles: “CDCL3” revise as “CDCl3”, The letter “L” is not capital

-          The NMR spectra of compound 3 show that the purity of this compound is very poor, and the authors should provide better quality/purity spectra?

-          HRMS and IR spectra of the new compounds 1-4 should be added to SI file

-          1H and 13C NMR spectra of the known compounds (5-10) should be also included in Supplementary file.

Reviewer 2 Report

Manuscript titled “Daphane Diterpenes from the Stelleropsis tianschanica with Their Anti-tumor Activity” reports the isolation and characterization of novel diterpenes. Their bioactivities as anticancer agents were also tested in vitro. Authors conclude that they are able to exert antitumor activities, which can support further use of Stelleropsis tianschanica as medicinal plants.

The work’s topic is interesting and relevant, since structure elucidation of new and potentially useful compounds may lead to further experimentation in various fields. There are some comments and suggestions for the authors:

1.       Please define the abbreviation “ECD” mentioned in the abstract.

2.       Also in the abstract, please include the data for cisplatin that is referenced here (line 20), and include its error.

3.       Since some compounds were previously known, while others were reported in the present work, please mention which were known and which are new in the legend of figure 1.

4.       Section 2.1 describes the compounds as white powders, did the authors calculate the yield of each compound? If available, please consider including this data on this general description.

5.       Please rename “Bioactive activity” to “Anti-tumor activity” (line 158) to homogenize the use of this term as mentioned in the title and in section 3.5.

6.       Line 160 mentions that the compounds were tested against human gastric cells, using cisplatin as control. Please briefly explain if cisplatin is a common agent used to treat this or similar types of malignancies (it can be assumed that it is, but this is not immediately clear). Furthermore, its mechanism of action and that of the compounds being tested should also be briefly mentioned in this section.

7.       The discussion presented in this section (2.2) should also be expanded, since it is too brief and does not include enough in-depth information to support the anticancer activities proposed by the authors. This is a significant point that should be addressed in order to provide more robust arguments to support the knowledge reported in the present work.

8.       Please consider illustrating the overall process described in section 3.3 (compound isolation and purification)

9.       The conclusion mentions IC50 values of less than 20 µg/mL, while the abstract mentions 8.8 μM. Please be consistent and mention this data using the same units, in order to avoid any unnecessary ambiguity.

10.   Finally, the manuscript requires a revision to improve the overall writing. Please consider enlisting the help of a native English-speaking colleague or professional editing service for this purpose.

Round 2

Reviewer 1 Report

The authors have considered/responded to all my recommended edits/comments in the revised version of the manuscript, and I recommend the manuscript to be published in the present form in Molecules.

Reviewer 2 Report

Manuscript titled “Daphane Diterpenes from the Stelleropsis tianschanica with Their Anti-tumor Activity” reports various analyses aimed at isolating and characterizing novel diterpenes, as well as determining their in vitro antitumor potential. This version of the manuscript was revised according to comments and suggestions made during an initial revision; those made by the present reviewer include: 

1.       Providing the full name of the abbreviation “ECD” in the abstract. This has been fixed as requested.

2.       Including numerical data for cisplatin. This data was added.

3.       Mentioning in the caption of figure 1 which compounds were already known, and which are new. Authors mention that this was explained in line 53, but this figure’s caption does not include this data. Please add it.

4.       Adding yield data for the compounds analyzed. This data was not available.

5.       Renaming the header of section 2.2 to be consistent with the title. The change was made as requested.

6.       Briefly mentioning the role of cisplatin as an antitumor agent and expanding the discussion of section 2.2. The discussion was extended as requested.

7.       Describing the isolation and purification of the compounds. The data is adequately presented.

8.       Homogenizing IC50 data to the same units. Units were homogenized as requested.

9.       Revising the overall writing of the manuscript. Authors enlisted a professional service to revise their manuscript.

According to the aforementioned changes, it is apparent that the authors adequately considered and responded to most issues. There are no additional comments, except specifying known and new compounds on Figure 1 (comment 3), although this change does not warrant an additional revision round.